# Investigation on the Printed CNT-Film-Based Electrochemical Sensor for Detection of Liquid Chemicals

**DOI:** 10.3390/s21155179

**Published:** 2021-07-30

**Authors:** Jaeha Noh, Sangsu An, Changhan Lee, Jiho Chang, Snagtae Lee, Moonjin Lee, Dongmin Seo

**Affiliations:** 1Major of Electronic Materials Engineering, Korea Maritime and Ocean University, Busan 49112, Korea; jaeha_noh@g.kmou.ac.kr (J.N.); sangsu_an@g.kmou.ac.kr (S.A.); dgks114@g.kmou.ac.kr (C.L.); 2Department of Offshore Plant Management, Korea Maritime and Ocean University, Busan 49112, Korea; sangtae@kmou.ac.kr; 3Maritime Safety and Environmental Research Division, Korea Research Institute of Ships and Ocean Engineering (KRISO), Daejeon 34103, Korea; moonjin.lee@kriso.re.kr (M.L.); dseo@kriso.re.kr (D.S.)

**Keywords:** carbon nanotubes (CNT), printed CNT film (PCF), electrochemical sensor, hazardous, noxious substances (HNS)

## Abstract

We studied electrochemical sensors using printed carbon nanotubes (CNT) film on a polyethylene telephtalate (PET) substrate. The mechanical stability of the printed CNT film (PCF) was confirmed by using bending and Scotch tape tests. In order to determine the optimum sensor structure, a resistance-type PCF sensor (R-type PCF sensor) and a comb-type PCF sensor (C-type PCF sensor) were fabricated and compared using a diluted NH_3_ droplet with various concentrations. The magnitude of response, response time, sensitivity, linearity, and limit of detection (LOD) were compared, and it was concluded that C-type PCF sensor has superior performance. In addition, the feasibility of PCF electrochemical sensor was investigated using 12 kinds of hazardous and noxious substances (HNS). The detection mechanism and selectivity of the PCF sensor are discussed.

## 1. Introduction

Research on the sensor for chemical substance detection has become more important in recent years due to growing interest in the environment. Various materials such as semiconductors [1], organic materials [2], and ceramics [3,4] are used for chemical substances sensors. Among them, carbon nanotubes (CNT) have many advantages in sensor applications such as large surface area, excellent mechanical strength, and high electrical conductivity. Kong [5] reported the first result that the resistance of single-walled carbon nanotubes (SWCNTs) changes when exposed to gas molecules. At present, CNT-based sensors have a variety of applications, for example, in environmental monitoring, food, agriculture, and biology fields [6]. Environmental monitoring applications include hazardous and noxious substance (HNS) detection sensors. Rushi et al. [7] reported a field effect transistor (FET)-type sensor using SWCNTs that reacts with gaseous hazardous and noxious substances (HNS) such as benzene (C_6_H_6_), toluene (C_7_H_8_), and xylenes (C_8_H_10_). They have succeeded in detecting those HNS with a concentration below the permissible exposure limit (PEL) provided by the Occupational Safety and Health Administration (OSHA). In addition, environmental sensors for detecting pH [8] and heavy metal ions [9] have been investigated. Despite these various efforts, research on an electrochemical sensor that can estimate the concentration of HNS in seawater is still insufficient. Note that a vast amount of HNS is transported through the ocean every year, and HNS spill accidents occur. Hence, sensors that can quickly identify liquid HNS mixed in seawater are greatly needed [10].

In this study, we study a liquid chemical sensor using a CNT film to detect the spilled HNS. We investigated the fabrication of CNT film, mechanical properties, and detection properties of chemical substances. The performance of a CNT sensor was evaluated with 12 kinds of HNSs. Finally, we considered the possibility of selective detection of HNS. In particular, responses of CNT-based sensors have been attributed to effects arising within the tubes (intra-CNT), effects arising at contact points between tubes (inter-CNT), or effects due to the contact between the tubes and the electrodes (Schottky barrier modulations) [6]. However, it is rather focused on the gas sensing mechanisms; hence, discussion about the liquid sensing mechanism of CNT-based sensors is required.

## 2. Materials and Methods

In this experiment, CNT film (thickness ~50 μm) was fabricated by applying the casting method [11]. Thick polyethylene telepthalate (PET) substrate 200 μm was used. Multi-walled carbon nanotubes (MWCNT) were ultrasonically dispersed using isopropanol (IPA) for 3 h. The PET substrate was cleaned for 10 min using distilled ionized (DI) water in an ultrasonic bath. We used both styrene butadiene rubber (SBR) and carboxymethyl cellulose (CMC) as binding materials for CNTs, which are well-known binders for carbon-based electrodes. Lim et al. [12] reported the effect of binders on the rheological properties of microstructure formation of a lithium-ion battery anode. They found that the SBR can affect the dispersion of the graphite particles, especially at a low CMC concentration. The mixing ratio of CNT:(SBR + CMC):acetylene black is 78:17:5 [13]. Here, acetylene black was added to improve the conductivity of the PCF. The PCF was dried in air at 90 °C for 2 h to remove binder and moisture [13]. The surface shape and thickness of the PCF were analyzed using an optical microscope and a field emission-scanning electron microscope (FE-SEM). Electrical properties were investigated using an I-V sourcemeter (Keithley 2400). The mechanical properties of the PCF were confirmed by bending and Scotch tape tests [14]. The resistance change is defined as [1 − (R/R_0_)] × 100%, where R_0_ is the reference resistance, R is the resistance in the bending state. We fabricated both R-type and C-type PCF sensors in order to determine the optimum sensor structure. The R-type PCF sensor operates based on the chemiresistance change, while the C-type PCF sensor operates based on the ion transfer between the adjacent electrodes. The sensor response (ΔR) was defined as the resistance ratio with and without stimulation (ΔR = R_s_/R_0_ × 100%, where the R_0_ is the reference resistance, R_s_ is the resistance in saturation state).

The response of the sensor was confirmed by using 12 kinds of HNS. The HNS used in the experiment and its characteristics are summarized in Table 1. The required minimum detection concentration was determined based on the reported PEL of each HNS [15]. Finally, the electrochemical detection mechanism and the selectivity of the PCF sensors were discussed.

## 3. Results and Discussion

### 3.1. Structure and Electrical Properties of the Fabricated PCF

Figure 1a is a picture of C-type PCF fabricated on a PET substrate. The thickness of the PCF is 50 μm. The width of the electrode is 2 mm, and the inter-electrode spacing is 1 mm. Figure 1b is the FE-SEM image of PCF surface. CNT bundles and binders can be seen, and it has a large surface area for sensing. Figure 1c shows the current–voltage characteristics of R-type PCF and C-type PCF. From the current-voltage measurement, the resistance of the R-type and C-type PCFs were 218 Ω and 330 Ω, respectively. Figure 1d shows the Raman spectra of PCFs. A 532 nm laser line was used, and a typical Raman spectrum with the well-known MWCNT related peaks at 1580 cm^−1^ (G-band) and 1336 cm^−1^ (D-band) were observed [17].

### 3.2. Mechanical Properties of PCF

Figure 2 shows the mechanical properties of PCF. Generally, bending and adhesion (so-called ‘Scotch tape test’) tests are carried out for the application of films on flexible substrates [18]. Figure 2a shows the resistance change with respect to the change of the bending radius. During the bending test within a radius from 0 to 30 mm in both inner and outer directions, the PCF shows a small resistance change (<4.3%). It is comparable with the previous result. Saran et al. [19] have reported a small resistance change of <3% from a bending test of 35 μm thick CNT film. Figure 2b,c reveal the before and after test FE-SEM images of PCF. Considerable change in morphology was not observed by the test. 

Figure 3a,b are the Scotch tape test [20] results. No pretreatment of the PET surface [21] to increase the adhesion of the film was performed. Along with the definition, the surface of PCF was scratched using a surgical blade. The scratched surface (Figure 3a) and the surface after attaching and removing the Scotch tape (Figure 3b) were observed by using an optical microscope. The adhesiveness of PCF was judged as excellent at level-4. It can be attributed to the structural properties of the CNT, which is known as the structural interlocking effect [22]. Zhang et al., found that the interfacial shearing strength of a composite film with an epoxy matrix and a CNT improves due to an interfacial interlocking effect of CNTs, which indicates the CNTs act as the core of nucleation. Similar effects were observed from Ag/CNT composite film [22] and carbon fiber/CNT film [23].

### 3.3. Temporal Response and Detection Mechanism of PCF Sensors

After investigating the mechanical strength of the PCF, the sensing properties of the PCF was observed. Figure 4 shows the response of each sensor. A chemical solution was applied, prepared by mixing salt-water (3.5wt % NaCl) and NH_3_ (99%, NH4OH). Note that OSHA has specified a PEL of NH_3_ as 25 ppm. Hence, we performed the experiment in the range of 1~50 ppm. Figure 4a,b show the response of both C-type and R-type PCF sensors, in which the inset of each figure is a photograph of the corresponding sensor. A resistance change according to the NH_3_ concentration was clearly observed. Figure 4c,d shows the concentration dependency of sensor response. From the results of Figure 4, sensor performances at room temperature such as sensitivity, LOD, response time, and linearity were summarized in Table 2, although the sensing performance of the PCF sensor was not optimized yet.

The experimentally confirmed LOD was 1 ppm for C-type and 10 ppm for R-type PCF sensor, while the LOD [24] of the sensors also can be defined theoretically by the following formula.
(1)$$\mathrm{Limit}\;\mathrm{of}\;\mathrm{Detection}=3\frac{S}{m}$$
where *S* is the standard deviation of the blank measurements and *m* is the slope of the calibration curve. By applying this formula, the LOD values of the C-type and R-type PCF sensors were calculated to be 0.011 ppm and 0.025 ppm. There were several reports about NH_3_ detection using CNT sensors. Huh et al., reported the detection of 5 ppm NH_3_ gas at 300 °C by a screen-printed SWCNT sensor [25]. They reported that saturation behavior was observed from 40 ppm. Wang et al., reported the detection of 5~200 ppm of NH_3_ at room temperature through MWCNT film produced by CVD (chemical vapor deposition) [26]. They found that the conductance of the sensors decreases when the sensors were exposed in NH_3_ due to the phase change of CNTs from metallic to semiconducting state. Note that both experimental and theoretical LOD values of those earlier experiments are comparable with the reported values in our experiment. For the purpose of comparison, it is worth mentioning that a tin dioxide-polyaniline nanocomposite flexible sensor revealed an ability to detect low NH_3_ concentrations of 10–200 ppb [27]. It is expected that improved performance can be obtained by optimizing device structure or/and introduction of catalysts.

The response time of the C-type PCF sensor at 25 ppm NH_3_ was 87 s, while it was 31 s for the R-type PCF sensor. Although C-type PCF sensor has a slow response time, it reveals a larger response, lower LOD level, and more linear response than the R-type PCF sensor. Therefore, we concluded that the C-type PCF sensor is more suitable for a liquid chemical sensor to detect HNS diluted in seawater.

The detection mechanism of the C-type PCF sensor is considered. The resistance of the C-type PCF can be explained by the charge transfer rate between electrodes. The charge transfer is determined by the mass transfer by diffusion and can be expressed by the following equation [28].
(2)$$i_{d}=\frac{{\mathrm{nFAD}}_{0}^{1/2}C_{0}}{\sqrt{\pi t}},$$
where i_d_ is the diffusion current, n is the number of reaction electrons, F is the Faraday constant, A is the area of the electrode, D_0_ is the diffusion coefficient, and C_0_ is the initial concentration of redox species. If the measurement principle of chronoamperometry is applicable to the results of this experiment, it will be proportional to C_0_ and decrease in proportion to t^−1/2^, which can be confirmed by plotting the Anson plot as shown in Figure 5. It can be interpreted that the response is determined in proportion to the concentration (C_0_) of HNS. Furthermore, the increase in slope at the high concentration range can be attributed to the structural effect of sensor.

### 3.4. Chemical Detection Properties of C-Type PCF Sensor

We have investigated the detection of various HNSs using the C-type PCF sensor. Twelve kinds of HNS were chosen as listed in Table 1. Figure 6 shows three selected results about the iso-propanol (IPA), ethanol (EtOH), and methanol (MeOH) which have PELs of 200 ppm [14]. We varied the concentration of each HNS from 1 to 400 ppm. Experimental LOD for EtOH was 10 ppm, while it was 1 ppm for MeOH and IPA. The other sensor properties such as theoretical LOD and the response time at the PEL are also summarized in Table 1. In this experiment, styrene showed the highest theoretical LOD value of 6.39 ppm. In addition, heptane revealed the slowest response time of 242 s. It is clearly shown that the C-type PCF sensor is very feasible for detecting various HNSs diluted in seawater.

It is well known that not only high sensitivity but also the selectivity of the sensor is an important factor for the development of sensors. The correlation between the response of the sensor and the physical properties of HNS has been considered, as shown in Figure 7. Figure 7a shows the correlation between the sensor responses and the polarity index of HNS. The coefficient of determination (R^2^) from a linear fitting was as low as 0.55. Figure 7b shows the correlation between response time and polarity index. In this case, R^2^ was 0.83, which indicates that response time is more strongly correlated with the polarity index. Moreover, note that non-polar HNS showed a longer response time (>140 s), while polar HNSs revealed a shorter response time (<110 s). It can be explained in terms of the difference in the diffusion coefficients of each HNS [29,30]. It is interesting to note that the risk index of non-polar HNS is generally higher than that of polar HNS [16]; hence, one could use the difference response time as a guide of warning high-risk HNS. Murad reported that external electric fields on the ion mobility, drift velocity, and drift–diffusion rate of ions. In our experiment, the external field was fixed to 1 V, and the effect of the field is much less dramatic in water than the ions; hence, ionic concentration will dominate the drift velocity and diffusion rate [31].

## 4. Conclusions

In this study, we have investigated the fabrication and performances of PCF-based electrochemical sensor. The mechanical and electrical properties of the PCF were evaluated. Furthermore, R-type and C-type PCF sensors were fabricated and compared to determine the optimum sensor. C-type PCF sensor was selected, and it was applied for the detection of 12 kinds HNSs. It successfully detected all HNSs at the concentrations below PEL. The possibility of selecting polar and non-polar HNS was confirmed. In conclusion, it was shown that the C-type PCF sensor is feasible as an electrochemical sensor for detecting diluted HNS in seawater.

## Figures and Tables

**Figure 1 sensors-21-05179-f001:**
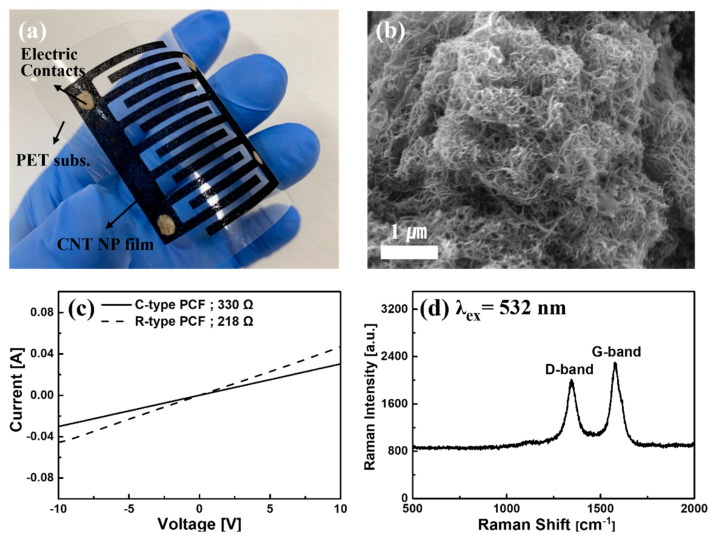
(**a**) CNT film printed on a PET substrate, (**b**) FE-SEM image of PCF surface, (**c**) I-V test results of C-Type and R-type PCFs, and (**d**) Raman spectra of PCF.

**Figure 2 sensors-21-05179-f002:**
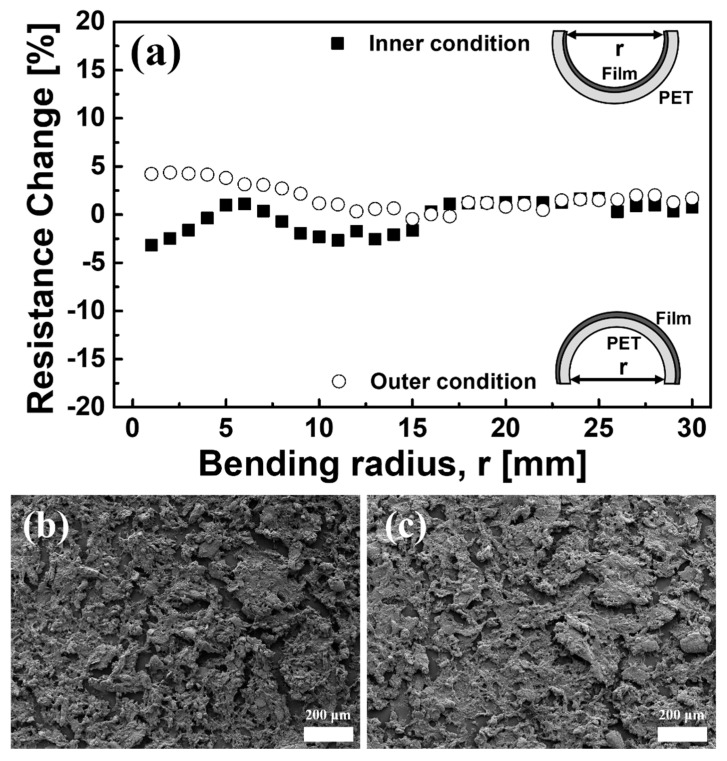
(**a**) Bending test result of PCF. The resistance change is calculated by: [1 − (R/R_0_)] × 100 (%); (**b**) SEM morphology of pristine PCF, (**c**) SEM morphology after bending test of PCF.

**Figure 3 sensors-21-05179-f003:**
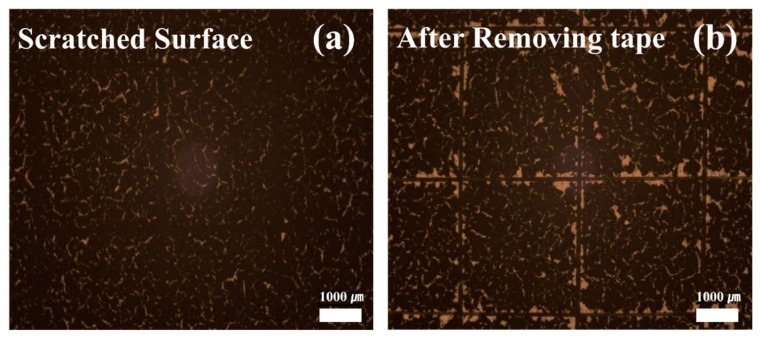
(**a**) Scratched surface of PCF and (**b**) PCF surface after removing the scotch tape. Note that the adhesion of PCF is evaluated through the peeled area of the surface.

**Figure 4 sensors-21-05179-f004:**
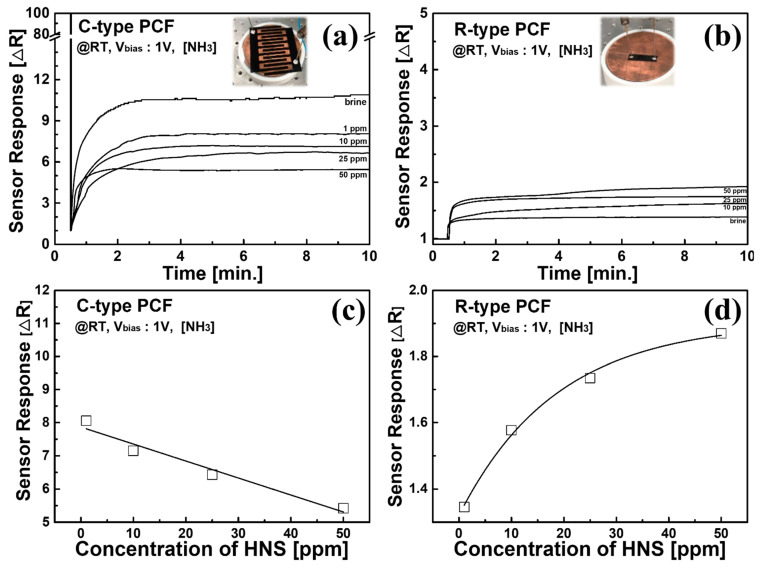
Temporal response of (**a**) C-type PCF and (**b**) R-type PCF sensors. Moreover, the sensitivity of (**c**) C-type and (**d**) R-type PCF sensors.

**Figure 5 sensors-21-05179-f005:**
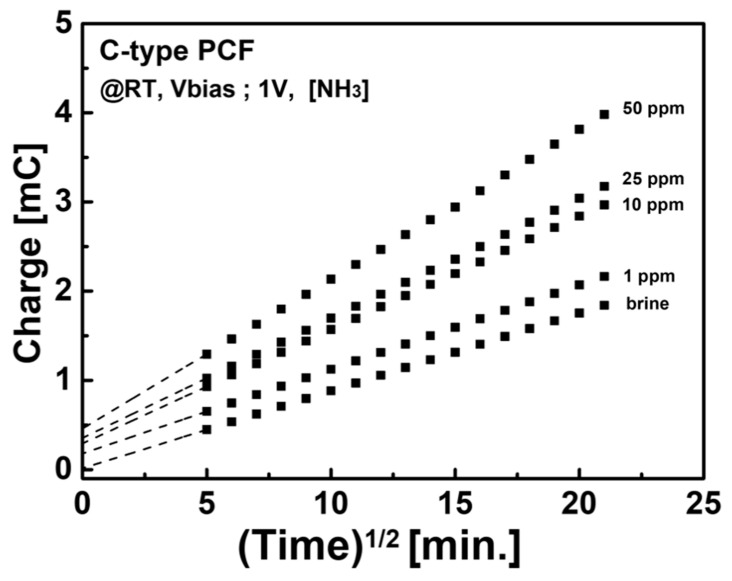
Anson plot of C-type PCF Sensor. Charge transfer corresponds with NH_3_ concentration.

**Figure 6 sensors-21-05179-f006:**
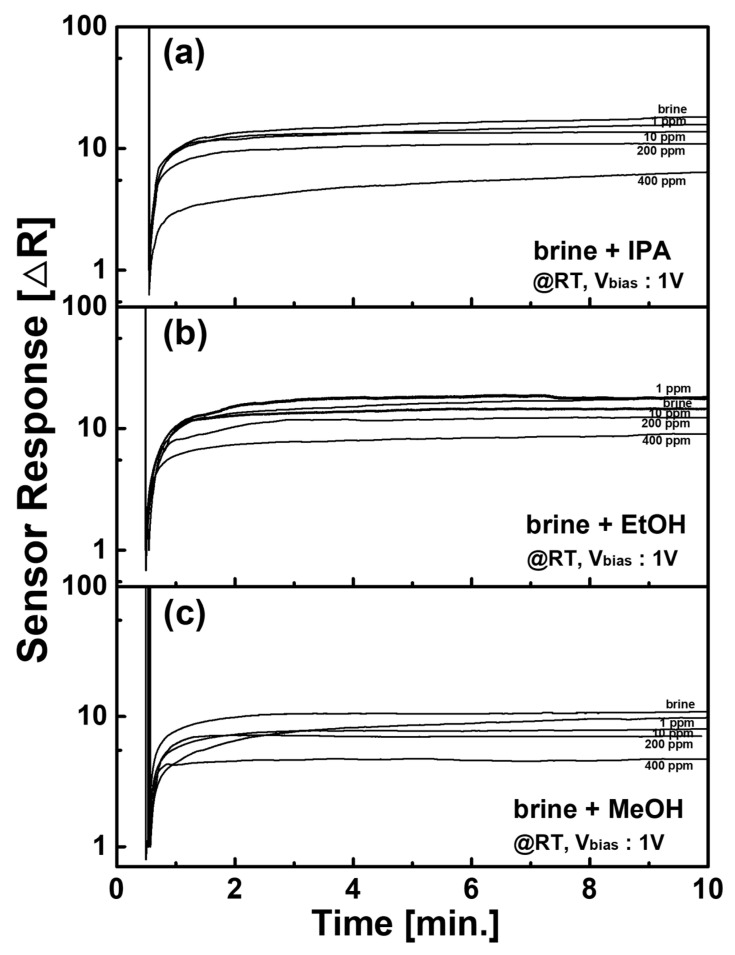
Temporal response of C-type PCF sensor against (**a**) IPA, (**b**) EtOH, and (**c**) MeOH in brine.

**Figure 7 sensors-21-05179-f007:**
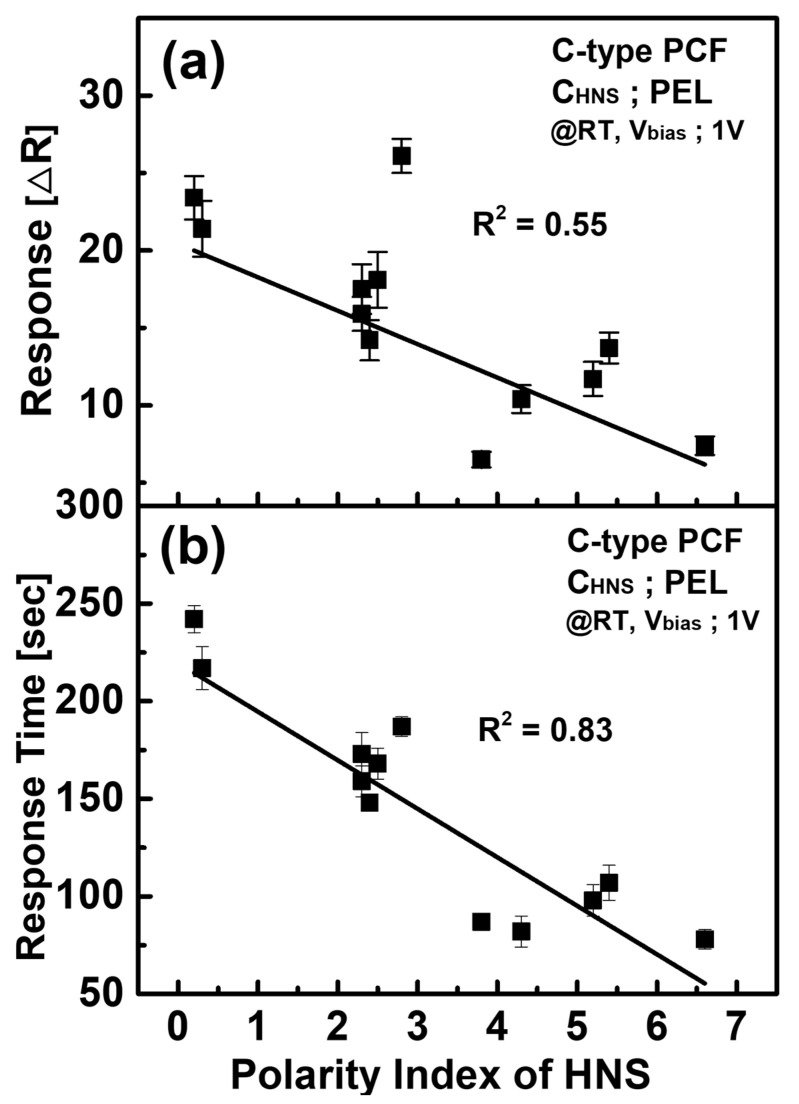
(**a**) Response and (**b**) response time characteristics of the C-type PCF sensor according to the polarity index.

**Table 1 sensors-21-05179-t001:** List of 12 kinds of HNS used in this experiment.

Polarity	Substances	Permeable Exposure Level *^1^ [ppm]	Polarity Index *^1^	Physical Property in Seawater/ Floating [F], Solubility [S], Evaporating [E]	Limit of Detection *^2^ [ppm]	Response Time *^2^ (Standard Error) [s]
Polar	Methyl alcohol	200	6.6	[F], [S]	0.54	78 (±5)
	Ethyl alcohol	200	5.2	[F], [S]	0.12	98 (±8)
	Iso-propanol	200	4.3	[F], [S], [E]	0.07	82 (±8)
	Acetone	500	5.4	[F], [S], [E]	2.49	107 (±9)
	Ammonia	25	3.8	[F], [S], [E]	0.01	87 (±4)
Non-Polar	Vinyl acetate	10	2.8	[F], [S], [E]	3.45	187 (±5)
	Benzene	0.5	2.3	[F], [E]	0.07	173 (±11)
	Ethyl Benzene	100	2.4	[F], [E]	1.53	148 (±4)
	Styrene	20	2.5	[F], [E]	6.39	168 (±8)
	Toluene	50	2.3	[F], [E]	0.22	159 (±8)
	Heptane	400	0.2	[F], [E]	1.06	242 (±7)
	n-Hexane	50	0.3	[F], [E]	10.9	217 (±11)

*^1^ reference [10,15,16]. *^2^ experimental results of this study.

**Table 2 sensors-21-05179-t002:** Sensor performances of both C-type and R-type PCF sensors.

Sensor Performances	C-Type PCF	R-Type PCF
Sensor Response, ΔR at 25 ppm (Standard Error)	6.43 (±0.29)	1.73 (±0.14)
Sensitivity (ΔR/ΔC_HNS_) [/ppm] (Standard Error)	0.053 (±0.003)	0.011 (±0.001)
Limit of Detection (LOD) [ppm]	0.011	0.025
Response Time, τ_R_ at 25 ppm [s] (Standard Error)	87 (±4)	31 (±1.4)
Coefficient of determination (R^2^) for a linear fitting	0.96	0.88

## Data Availability

The data presented in this study are available on request from the corresponding author. The data are not publicly available due to privacy restrictions.

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
