# Peer review of "Investigation on the Printed CNT-Film-Based Electrochemical Sensor for Detection of Liquid Chemicals"

_sensors, 2021, doi:10.3390/s21155179_

Round 1

Reviewer 1 Report

The authors designed two electrochemical sensors based on printed carbon nanotube films (PCF) for the detection of various hazardous substances. They have characterized in detail the surface morphologies, mechanical and electrical properties of the fabricated sensors. The C-type PCF showed more superior performance than the R-type sensor in terms of lower detection limit and more linear response. The selectivity, sensitivity, and detection mechanism of the C-type PCF sensor were presented and discussed in a very logical manner. I would recommend it for publication in Sensors after addressing a small issue listed below.

1). Caption of Figure 2(c) was missing. (See Page 4, Line 107)

Author Response

Reviewer

Sensors

Subj. Response letter

Title : Investigation on the printed CNT film based Electrochemical Sensor

Corresponding Author : Jiho Chang

On behalf of all authors I would like to appreciate on your valuable comments.

Reviewer #1 ’s comment:

1) Caption of Figure 2(c) was missing. (See Page 4, Line 107)

Revised figure caption : Figure 2. (a) bending test result of PCF. The resistance change is calculated by; [1-(R/R0)]×100 (%). (b) SEM morphology of the pristine PCF, (c) SEM morphology after bending test of PCF.

Reviewer 2 Report

This communication reports the use of a printed CNT film as a sensor for HNS. This paper shows some preliminary results showing some potential of the sensor. However, some improvements must be done in this short communication:

  • The title should include the application of the sensor
  • In the abstract and main text the abbreviations must be previously defined.
  • In Line 56 “50 mm” must be corrected.
  • In Line 112 “NH3OH” must be corrected.
  • The plots of the experimental data must have error bars or an error band – using the same PET device and different PET devices.
  • Experimental results must have an error (Tables 1 and 2, and in the text).
  • In Fig. 4 the units of the axis are missing.
  • Apparently some of the non-polar HNS have lower water solubility than 400 ppm! How were the solutions prepared? The seawater solubility must be shown.

Author Response

Reviewer

Sensors

Subj. Response letter

Title : Investigation on the printed CNT film based Electrochemical Sensor

Corresponding Author : Jiho Chang

On behalf of all authors I would like to appreciate on your valuable comments.

For reviewer #2, the revised sentences in manuscript were indicated by blue color.

Reviewer #2’s comment :

1) The title should include the application of the sensor

→ Title of this manuscript is changed as “Investigation on the Printed CNT Film Based Electrochemical Sensor for Detection of Liquid Chemicals”. See page 1, line 2

2) In the abstract and main text the abbreviations must be previously defined.

→ The abbreviations in the abstract are defined. See page 1, line 14 (CNT), line 15 (PET), line 15 (PCF), and line 19-20 (LOD). Also it was revised in the introduction section too. See page 1, line 34 (SWCNT), line 38 (FET), line 41 (PEL), and line 42 (OSHA).

3) In Line 56 “50 mm” must be corrected.

→ The unit of thickness “mm” is revised to “μm”. See page 2, line 64

4) In Line 112 “NH3OH” must be corrected.

→ The chemical formula “NH3OH” is revised to “NH3OH”. See page 5, line 137

5) The plots of the experimental data must have error bars or an error band – using the same PET device and different PET devices.

→ We put the error bars for the experimental data in the Fig. 7.

6) Experimental results must have an error (Tables 1 and 2, and in the text).

→ Table 1 and 2 are revised including standard errors for the experimental data. 

7) In Fig. 4 the units of the axis are missing.

→ Fig. 5 is revised. As shown in the Fig. 5, units for horizontal and vertical axis are revised as [minutes] and [coulomb], respectively.

8) Apparently some of the non-polar HNS have lower water solubility than 400 ppm! How were the solutions prepared? The seawater solubility must be shown.

→ Permeable exposure level (PEL) of both soluble and insoluble HNS is defined by the administrated amount of HNS per 1L of seawater, nevertheless various transformations such as evaporation and dissolution will happen in the real situation at sea.

→ Table A shows the solubility of various HNSs in pure water. In general, hydrocarbons are less soluble in seawater than distilled water, and the solubility is influenced by the salting-out (especially for the molecules with heavy molar weight) and temperature. [Ref. #2]

Table A. Solubility of solutions in pure water [Ref. #1, Ref. #2]

HNS

Solubility

(in pure Water, mg/L)

HNS

Solubility

(in pure Water, mg/L)

Methanol

1,000,000

Benzene

1,780

Ethanol

579,000

Ethyl Benzene

152

IPA

1,000,000

Styrene

300

Acetone

193,000

Toluene

515

Ammonia

346,000

Heptane

8.9

Vinyl acetate

65,300

n-Hexane

33

[References]

[#1] French McCay, D. P.; Whittier, N.; Ward, M.; Santos, C. (2006). Spill hazard evaluation for chemicals shipped in bulk using modeling. Environ. Model. Softw., 2006, 21(2), 156–169. doi:10.1016/j.envsoft.2004.04.021 

[#2] Rossi, S. S., & Thomas, W. H. (1981). Solubility behavior of three aromatic hydrocarbons in distilled water and natural seawater. Environmental Science & Technology, 15(6), 715–716. doi:10.1021/es00088a013

Reviewer 3 Report

From my point of view, the publication of the communication (Investigation on the printed CNT film-based electrochemical sensor) can only be considered after major revisions.

The introduction part should be improved and the author would be discussed a little bit more previous references about HNS detection mechanism SWCNT or CNT.

line no 60 “SBR (styrene butadiene rubber) and CMC (carboxymethyl cellulose) binders were used.” is not clear. Please re-write this line.

Line no 66 should re-write and mention the bending parameter.

How many samples are measured for this study? Please use the average value with deviation.

Include SEM images of before and after bending experiments.

The result and discussion part need to improve especially just not put experimental value but also explain the results.

References must be the same style. Ref-13 (line 237-39) style does not match with others

Author Response

Reviewer

Sensors

Subj. Response letter

Title : Investigation on the printed CNT film based Electrochemical Sensor

Corresponding Author : Jiho Chang

On behalf of all authors I would like to appreciate on your valuable comments.

For reviewer #3, the revised sentences in manuscript were indicated by Green color.

Reviewer #3’s comment :

1) The introduction part should be improved and the author would be discussed a little bit more previous references about HNS detection mechanism SWCNT or CNT.

→ The detection mechanism of CNT based sensors was summarized well by Schroeder et al [Chemical reviews 2019 119, 599-663]. It is noticed at the end of introduction part as; “Especially, responses of CNT-based sensors are attributed to effects arising within the tubes (intra-CNT), effects arising at contact points between tubes (inter-CNT), or effects due to the contact between the tubes and the electrodes (Schottky barrier modulations) [6]. However, it is rather focused on the gas sensing mechanism, hence discussion about the liquid sensing mechanism of CNT based sensors are required.”

See page 2, line 53-58

2) line no 60 “SBR (styrene butadiene rubber) and CMC (carboxymethyl cellulose) binders were used.” is not clear. Please re-write this line.

→ Line 60 is changed as “We used both styrene butadiene rubber (SBR) and carboxymethyl cellulose (CMC) as binding materials for CNTs. Actually, those are well known binders for carbon-based electrodes. Lim et al [12] reported the effect of binders on the rheological properties of microstructure formation of lithium-ion battery anode. They found that the SBR can affect the dispersion of the graphite particles especially at low CMC concentration.” See page 2, line 68-73

3) Line no 66 should re-write and mention the bending parameter.

→ A description of the bending parameter was added. See page 3 line 79-81

“The resistance change is defined as [1-(R/R0)]×100%, where R0 is the reference resistance, R is the resistance in the bending state”

4) How many samples are measured for this study? Please use the average value with deviation.

→ Three or more samples are used to obtain one measurement result. Standard errors were included in the experimental results. See Table 1, 2 and Figure 7.

5) Include SEM images of before and after bending experiments.

→ The FE-SEM images of before and after the bending experiments were included in Figure 2 (b) and (c). Considerable surface change was not observed.

6) The result and discussion part need to improve especially just not put experimental value but also explain the results.

→ Additional discussion is added on the interlocking effect according to the bending test results. See page 4, line 126-130

→ Additional discussion on the results related to the sensor characteristics is added. See page 5, line 153-163

→ Additional consideration on the response time of the PCF sensor is added. See page 8, line 215-218

7) References must be the same style. Ref-13 (line 237-39) style does not match with others

→ It was revised along with the guideline of the journal.

→ The Ref. 13 is changed as [Ref. 14] Lee, S. W.; Kim, K. K.; Cui, Y.; Lim, S. C.; Cho, Y. W.; Kim, S. M.; Lee, Y. H. Adhesion test of Carbon Nanotube Film coated onto transparent conducting substrates. Nano, 2010, 05(03), 133–138. doi:10.1142/s1793292010002025. See page 10, line 274-275

Reviewer 4 Report

The paper discusses the fabrication and sensing capabilities of PCF based sensors for 12 kinds hazardous and noxious substances. The studies and analysis are pretty thorough and interesting. The overall presentation of the experiments and results are well organized and interesting. I'd recommend publish in present form.

Author Response

Reviewer

Sensors

Subj. Response letter

Title : Investigation on the printed CNT film based Electrochemical Sensor

Corresponding Author : Jiho Chang

Reviewer #4’s comment :

The paper discusses the fabrication and sensing capabilities of PCF based sensors for 12 kinds hazardous and noxious substances. The studies and analysis are pretty thorough and interesting. The overall presentation of the experiments and results are well organized and interesting. I'd recommend publish in present form.

→ On behalf of all authors I would like to appreciate on your valuable comments.

Round 2

Reviewer 2 Report

The authors have satisfactorily answered the referee suggestions. Only one mistake is still present, ammonium hydroxide is NH4OH and should be corrected. 

Reviewer 3 Report

I believe the manuscript has been sufficiently improved to warrant publication in Sensors.